# Modelling the impact of interventions on the progress of the COVID-19 outbreak including age segregation

**Jorge Rodríguez**[1]*, **Mauricio Patón**[1], **Joao M. Uratani**[1], **Juan M. Acuña**[2]

**1** Department of Chemical Engineering, College of Engineering, Khalifa University, Abu Dhabi, United Arab Emirates, **2** Department of Epidemiology and Public Health, College of Medicine, Khalifa University, Abu Dhabi, United Arab Emirates

* jorge.rodriguez@ku.ac.ae

## Abstract

In this work, a SEIR-type mathematical model of the COVID-19 outbreak was developed that describes individuals in compartments by infection stage and age group. The model assumes a close well-mixed community with no migrations. Infection rates and clinical and epidemiological information govern the transitions between stages of the disease. The impact of specific interventions (including the availability of critical care) on the outbreak time course, the number of cases and the outcome of fatalities were evaluated. Data available from the COVID-19 outbreak from Spain as of mid-May 2020 was used. Key findings in our model simulation results indicate that (i) universal social isolation measures appear effective in reducing total fatalities only if they are strict and the number of daily interpersonal contacts is reduced to very low numbers; (ii) selective isolation of only the elderly (at higher fatality risk) appears almost as effective as universal isolation in reducing total fatalities but at a possible lower economic and social impact; (iii) an increase in the number of critical care capacity directly avoids fatalities; (iv) the use of personal protective equipment (PPE) appears to be effective to dramatically reduce total fatalities when adopted extensively and to a high degree; (v) extensive random testing of the population for more complete infection recognition (accompanied by subsequent self-isolation of infected aware individuals) can dramatically reduce the total fatalities only above a high percentage threshold that may not be practically feasible.

## Introduction

COVID-19 is the worst pandemic the world has seen in a century. The devastating consequences of the pandemic, both in number of fatalities as in economic harm, have brought unprecedented attention from nearly all fields. The large number of cases in a short period of time have forced governments to implement interventions to slow the spread of the virus. Such interventions consisted of the isolation of individuals (e.g. via curfews or full lockdowns, the banning of people in public events (e.g. concerts, stadiums), the enforcement to use

**Data Availability Statement:** The Matlab® source code and Excel file containing all parameter values used as well as a non-age segregated version of

the model are available at https://github.com/EnvBioProM/COVID_Model.

**Funding:** This study was supported by Khalifa University in the form of a grant awarded to JR (8474000317 CRPA-2020-SEHA).

**Competing interests:** The authors have declared that no competing interests exist.

protective equipment masks in public places, the building of field hospitals and the implementation of mass testing and contact tracing among others.

Epidemic modelling has a long history [1–9], as the prediction of the course and possible outcomes of a pandemic is crucial for society. The field has flourished during the pandemic period, with several mathematical models for the COVID-19 disease published from March 2020 to date. Some of these modelling efforts have evaluated the evolution of the disease spread in several countries such as in Italy [10, 11], France [12], Australia [13], the United Kingdom, the United States [14, 15], Ukraine [16], India [17] or Africa [18]. Other modelling efforts have focused on the earlier dynamics of transmission from initial cases, to the potential of the implementation of interventions to limit the disease spread, such as international travel restrictions [19, 20], contact tracing and isolation of infected individuals at onset [21], the use of masks [22], different scales of social distancing and isolation or the impact on the healthcare system [23].

Other works have focused on the estimation and/or the understanding of fundamental characteristics (i.e. potential model parameters) of the disease, such as the incubation period [24], how the transmission might occur, as well as to assess short-term [25] and long term forecasts [26].

In this work a modified SEIR-type model that evaluates intervention scenarios for the COVID-19 outbreak is presented. The model considers an approach based on the segregation by age groups and known disease stages. The model aims at retaining mechanistic meaning of all variables and parameters while capturing the relevant phenomena at play. The model aims also at contributing to the understanding by broader audiences (researchers, public health authorities, and the general public) of what to expect on the propagation of infectious diseases and how specific interventions may help. The complexity level was intentionally limited to maintain accessibility to non-experts and policy makers to comprehend the model results such that expert advice and decision making can be brought closer together to help guide interventions for immediate and longer-term needs.

## Materials and methods

### Model description

The model presented is based on balances of individuals, segregated by age group, transitioning between infection stages. All individuals are placed in a common single domain or compartment (e.g. a well-mixed city or town), no geographical clustering nor separation of any type is considered, nor is any form of migration in or out of the community. Large cities with ample use of public transportation are thought to be settings best described by the model.

The model also provides a direct estimation of the effective reproduction number($R_t$) [27, 28] under different circumstances of individual characteristics (such as use of personal protection or awareness) as well as under population-based interventions (such as imposed social isolation). The basic reproduction number ($R_0$) provides an estimation of the number of cases expected to be generated by one case in the population when all individuals are considered to be susceptible [28, 29]. $R_t$ however, provides the average number of secondary infections per primary case at a given time [30] and it is used to evaluate the effectiveness of interventions [31]. The ability to estimate the $R_t$ for different times of the outbreak (given the interventions), outbreak settings and interventions is considered to be a valuable characteristic of the model. $R_t$ is predicted to change over time with interventions not related to increased immunity (isolation or use of personal protection equipment (PPE), as opposed to vaccination).

**Model constituents.** The model solves dynamic variables or states. Every individual belongs, in addition to their age group (which she/he never leaves), to only one of the possible

stages of the infection in terms of infectiousness and severity of symptoms, namely: *healthy susceptible* (H); *asymptomatic non-infectious* (NI); *asymptomatic infectious* (AS); *symptomatic* (S); *in need of hospitalisation* (SH); *in need of critical care* (SC); *deceased* (D) and *recovered immune* (R).

Definitions of the model state variables are shown in Table 1. Each variable is a state vector with the number of individuals in that stage per age group, which is defined per decade from 0–9 to 80+ year olds (9 age groups). Therefore, each state is a vector of dimensions 1x9, and the total number of states is a matrix of dimensions 8x9. Note that vector variables and parameters are represented in bold font and scalar ones in regular font.

Fig 1 shows the population progress through stages as described in the model. An additional schematic representation of the model approach with the population groups considered for the infection stages, rates of infection and transition between groups and showing possible interactions between population groups is shown in S5 Fig in S1 File.

**Rates of transition between infection stages.**   The transitions between stages are governed by the rates of infection and transition shown in Table 2. Notation is defined as $\mathbf{r_{i,j}}$, where the individual moves from the compartment *j* to the compartment *i*.

The average rates of transition between states are defined such that the latest epidemiological and clinical data can be used to determine and continuously update the parameters as more knowledge of the disease emerges. These parameters include the proportion of individuals that transition to a more severe stage or recover (see Table 3) and the average times reported at each stage before transition or recovery (see Table 4).

The rates of transition between stages healthy to critical (in number of individuals per day) are described in Eqs 1.A–1.E. All rates are vectors per age group of dimensions (1x9). Note that point operators between vectors indicate an operation element-by-element.

$$\mathbf{r_{ni\_h}} = \mathbf{r_{i\_as}} + \mathbf{r_{i\_s}} \tag{1.A}$$

$$\mathbf{r_{as\_ni}} = (\mathbf{f_{as\_ni}} . / \mathbf{t_{as\_ni}}) . * \mathbf{N_{ni}} \tag{1.B}$$

$$\mathbf{r_{s\_as}} = (\mathbf{f_{s\_as}} . / \mathbf{t_{s\_as}}) . * \mathbf{N_{as}} \tag{1.C}$$

$$\mathbf{r_{sh\_s}} = (\mathbf{f_{sh\_s}} . / \mathbf{t_{sh\_s}}) . * \mathbf{N_{s}} \tag{1.D}$$

$$\mathbf{r_{sc\_sh}} = (\mathbf{f_{sc\_sh}} . / \mathbf{t_{sc\_sh}}) . * \mathbf{N_{sh}} \tag{1.E}$$

**Table 1. Model states in vectors (1x9) of number of individuals in each infection stage.**

| Definition "Number of individuals…" | Variable | Totals of all ages |
|---|---|---|
| Healthy susceptible to infection | $\mathbf{N_h}$ | $N_{hT}$ |
| Non-infectious asymptomatic | $\mathbf{N_{ni}}$ | $N_{niT}$ |
| Infectious asymptomatic | $\mathbf{N_{as}}$ | $N_{asT}$ |
| Infectious symptomatic | $\mathbf{N_s}$ | $N_{sT}$ |
| Requiring hospitalisation | $\mathbf{N_{sh}}$ | $N_{shT}$ |
| Requiring critical care | $\mathbf{N_{sc}}$ | $N_{scT}$ |
| Deceased | $\mathbf{N_d}$ | $N_{dT}$ |
| Recovered & immune | $\mathbf{N_r}$ | $N_{rT}$ |

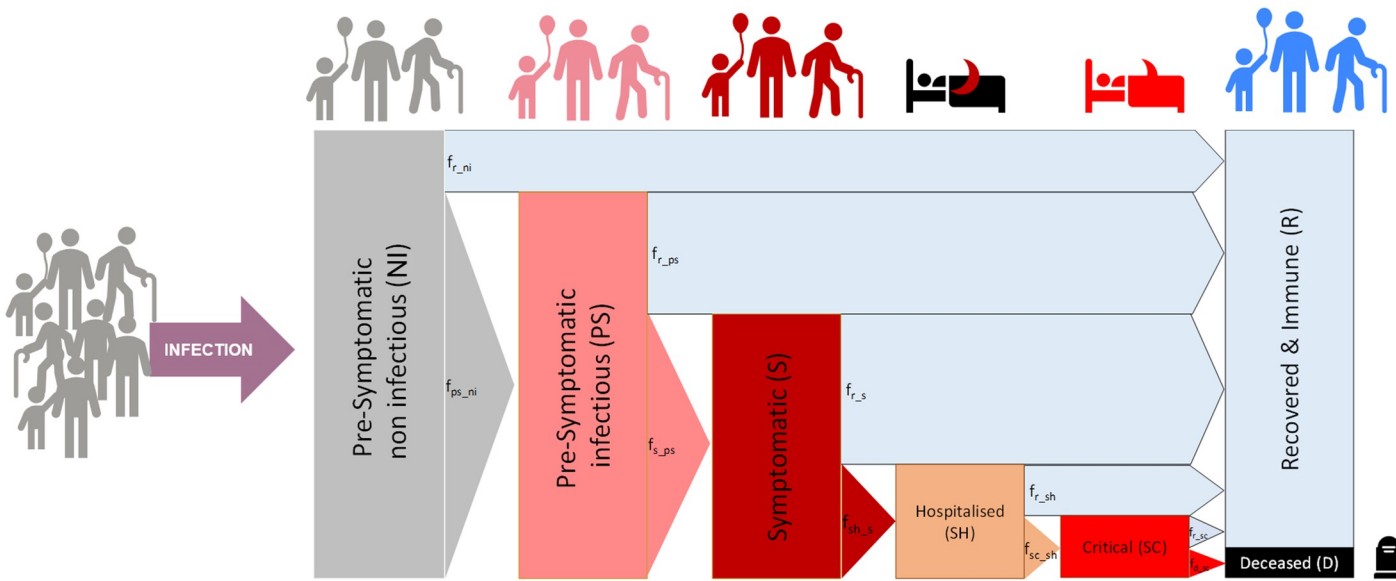

**Fig 1. Schematic representation of the population compartments by infection stages and age.** Only interactions between infectious individuals (both asymptomatic and symptomatic) and healthy susceptible ones can increase the rate of infections.

The rates of individuals recovering from the different infected stages (in number of individuals per day) are described in Eqs 2.A–2.E. (all rates in vectors per age group).

$$\mathbf{r_{r\_ni}} = (\mathbf{f_{r\_ni}}./\mathbf{t_{r\_ni}}).* \mathbf{N_{ni}} \tag{2.A}$$

$$\mathbf{r_{r\_as}} = (\mathbf{f_{r\_as.}}/\mathbf{t_{r\_as}}).* \mathbf{N_{as}} \tag{2.B}$$

$$\mathbf{r_{r\_s}} = (\mathbf{f_{r\_s}}./\mathbf{t_{r\_s}}).* \mathbf{N_{s}} \tag{2.C}$$

$$\mathbf{r_{r\_sh}} = (\mathbf{f_{r\_sh}}./\mathbf{t_{r\_sh}}).* \mathbf{N_{sh}} \tag{2.D}$$

$$\mathbf{r_{r\_sc}} = (\mathbf{f_{r\_sc}}./\mathbf{t_{r\_sc}}).* \mathbf{N_{sc\_ic}} \tag{2.E}$$

**Table 2. Rates of infection and transition between states, vectors (1x9).**

| Definition "Rate of …" | Variable | Units |
|---|---|---|
| Infection by interaction with infectious asymptomatic | $\mathbf{r_{i\_as}}$ | # H infected/day |
| Infection by interaction with infectious symptomatic | $\mathbf{r_{i\_s}}$ | # H infected/day |
| Transition from non-infectious to infectious asymptomatic | $\mathbf{r_{as\_ni}}$ | # NI to AS / day |
| Transition from asymptomatic to symptomatic | $\mathbf{r_{s\_as}}$ | # AS to S / day |
| Transition from symptomatic to hospitalised | $\mathbf{r_{sh\_s}}$ | # S to SH / day |
| Transition from hospitalised to critical | $\mathbf{r_{sc\_sh}}$ | # SH to SC / day |
| Transition from critical to deceased | $\mathbf{r_{d\_sc}}$ | # SC to D / day |
| Recovery from asymptomatic non-infectious | $\mathbf{r_{r\_ni}}$ | # NI to R / day |
| Recovery from asymptomatic infectious | $\mathbf{r_{r\_as}}$ | # AS to R / day |
| Recovery from symptomatic | $\mathbf{r_{r\_s}}$ | # S to R / day |
| Recovery from hospitalised | $\mathbf{r_{r\_sh}}$ | # SH to R / day |
| Recovery from critical | $\mathbf{r_{r\_sc}}$ | # SC to R / day |

**Table 3. Epidemiological parameters (all in vectors per age group).**

| Definition | Parameter | Units |
|---|---|---|
| Fraction of NI that will become AS | $f_{as\_ni}$ | #AS/#NI |
| Fraction of AS that will become S | $f_{s\_as}$ | #S/#AS |
| Fraction of S that will become SH | $f_{sh\_s}$ | #SH/#S |
| Fraction of SH that will become SC | $f_{sc\_sh}$ | #SC/#SH |
| Fraction of cared SC that will die into D | $f_{d\_sc}$ | #D/#SC$_{IC}$ |
| Fraction of NI that will recover into R[1] (1- $f_{as\_ni}$) | $f_{r\_ni}$ | #R/#NI |
| Fraction of AS that will recover into R[1] (1- $f_{s\_as}$) | $f_{r\_as}$ | #R/#AS |
| Fraction of S that will recover into R[1] (1- $f_{sh\_s}$) | $f_{r\_s}$ | #R/#S |
| Fraction of SH that will recover into R[1] (1- $f_{sc\_sh}$) | $f_{r\_sh}$ | #R/#SH |
| Fraction of cared SC that will recover into R[1] (1- $f_{d\_sc}$) | $f_{r\_sc}$ | #R/#SC$_{IC}$ |

[1]Calculated by difference with the complementary, not an input parameter.

The rate of transition from critical to deceased is the sum of that of those critical receiving intensive care ($r_{d\_scic}$) plus that of those critical without available care ($r_{d\_scnc}$) as per Eqs 3.A–3.C. All critical individuals not receiving intensive care ($N_{sc\_ncc}$) are assumed to become fatalities after a time ($t_{d\_nc}$). The allocation of critical care is described below.

$$r_{d\_sc} = r_{d\_scic} + r_{d\_scnc} \tag{3.A}$$

$$\text{where} \qquad r_{d\_scic} = (f_{d\_sc}./t_{d\_sc}).* N_{sc\_ic} \tag{3.B}$$

$$r_{d\_scnc} = (1./t_{d\_nc}).* N_{sc\_ncc} \tag{3.C}$$

Fig 2 provides a representation of the transition between stages, the proportions of individuals recovering, or worsening are as per Fig 1.

**Rates of infection.** The infection of healthy susceptible individuals (H) is modelled as occurring only via their interaction with infectious either asymptomatic (AS) or symptomatic (S) individuals. Hospitalised (SH) and critical (SC) individuals are assumed not available for interactions neither are those deceased (D).

Two rates of infection of healthy susceptible individuals (in number of infections per day) are defined, one from each one of the two possible infecting groups (AS and S). The rates of

**Table 4. Clinical average times in each infection stage (all in vectors per age group).**

| Definition | Parameter | Units |
|---|---|---|
| Time to become infectious | $t_{as\_ni}$ | days |
| Time to develop symptoms from infectious | $t_{s\_as}$ | days |
| Time to require hospitalisation from symptoms onset | $t_{sh\_s}$ | days |
| Time to require critical care from hospitalisation | $t_{sc\_sh}$ | days |
| Time to death from critical | $t_{d\_sc}$ | days |
| Time to death from critical with no care available | $t_{d\_nc}$ | days |
| Time to recover from asymptomatic non-infectious | $t_{r\_ni}$ | days |
| Time to recover from asymptomatic infectious | $t_{r\_as}$ | days |
| Time to recover from (non-severe) symptoms | $t_{r\_s}$ | days |
| Time to recover from hospitalisation | $t_{r\_sh}$ | days |
| Time to recover from critical | $t_{r\_sc}$ | days |

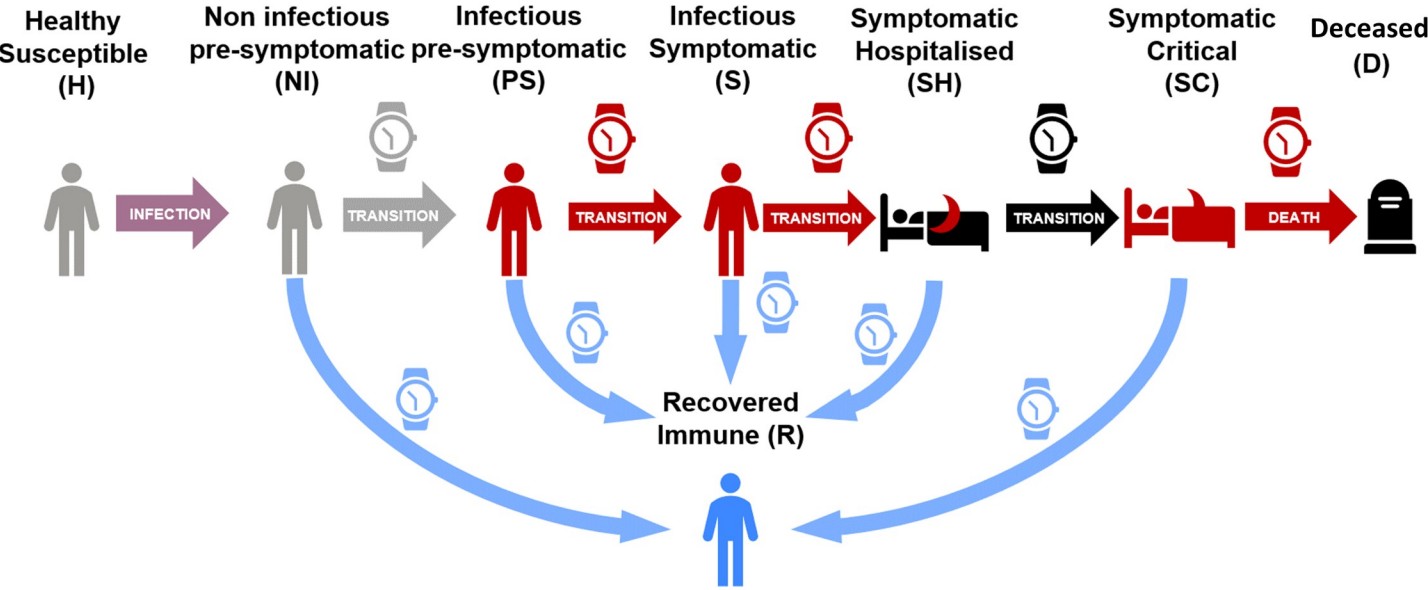

**Fig 2. Schematic representation of the transitions rates between stages.** Individuals spend in each stage an amount of time depending on their transition path towards recovery or increased severity.

infection (in vectors per age group) result from the product of (i) the fraction of interactions occurring with AS (or S) individuals among the total interactions ($f_{ias}$ or $f_{is}$) (Eqs 8.E and 8.F) times (ii) the likelihood of contagion in an interaction with AS (or S) ($\mathbf{p_{i\_as}}$ or $\mathbf{p_{i\_s}}$, Eqs 8.A and 8.B) (per age group), (iii) the average number of daily interpersonal contacts that H individuals have ($\mathbf{ni_h}$) and (iv) the number of H individuals themselves (per age group) (see Fig 3 and Eqs 4.A and 4.B).

$$\mathbf{r_{i\_as}} = \mathbf{N_h} .* \mathbf{ni_h} * f_{ias} .* \mathbf{p_{i\_as}}; \tag{4.A}$$

$$\mathbf{r_{i\_s}} = \mathbf{N_h} .* \mathbf{ni_h} * f_{is} .* \mathbf{p_{i\_s}}; \tag{4.B}$$

**Stage transition equations.** The dynamic variation on the number of individuals in each stage over time and per age group is governed by the population balance equations described

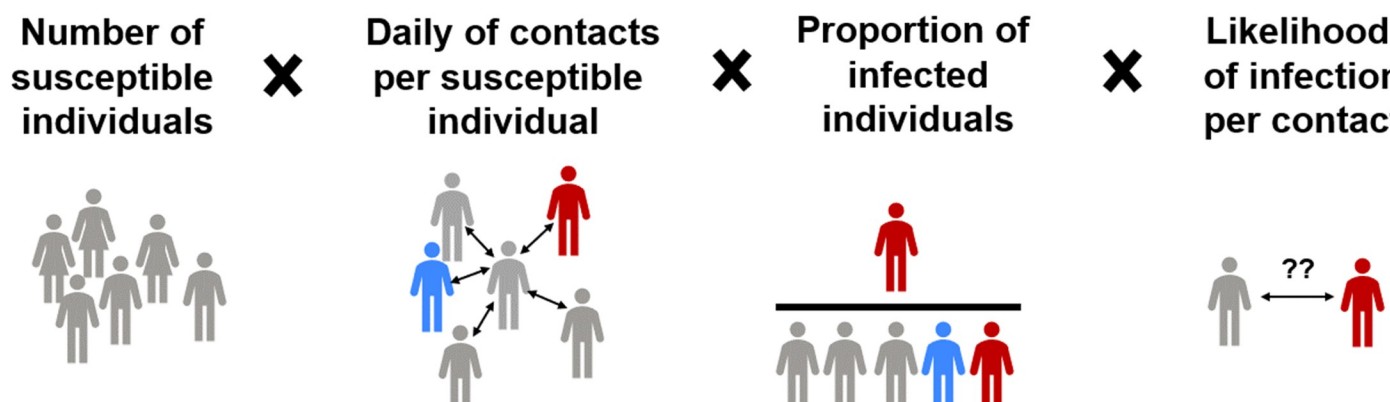

**Fig 3. Schematic representation of the terms of the infection rate.**

in Eqs 5.A–5.H. (all in vectors by age group).

$$dN_h/dt = -r_{ni\_h} \tag{5.A}$$

$$dN_{ni}/dt = r_{ni\_h}. - r_{as\_ni}. - r_{r\_ni} \tag{5.B}$$

$$dN_{as}/dt = r_{as\_h}. - r_{s\_as}. - r_{r\_as} \tag{5.C}$$

$$dN_s/dt = r_{s\_as}. - r_{sh\_s}. - r_{r\_s} \tag{5.D}$$

$$dN_{sh}/dt = r_{sh\_s}. - r_{sc\_sh}. - r_{r\_sh} \tag{5.E}$$

$$dN_{sc}/dt = r_{sc\_sh}. - r_{d\_sc}. - r_{r\_sc} \tag{5.F}$$

$$dN_d/dt = r_{d\_sc} \tag{5.G}$$

$$dN_r/dt = r_{r\_as}. + r_{r\_s}. + r_{r\_sh}. + r_{r\_sc} \tag{5.H}$$

The state transitions as governed by these rates are represented in a matrix form in S5a Fig in S1 File.

**Calculation of the effective reproduction number ($R_t$).**   The reproduction number refers to the potential infection of susceptible individuals from infected individuals [29]. Since the model produces, for each parameter set used, a set of values for its outputs over at any given time, an instantaneous estimation of the reproduction number ($R_t$) is also obtained. Several parameters and variables influence the $R_t$ including the duration of infectious stages; the likelihoods of infection per social contact as well as the percentages of individuals transitioning to more severe stages.

The effective reproduction number ($R_t$) during the outbreak is computed over time according to Eq 6 from the current values of the model state variables. Under this approach, infectious individuals can only infect others while they are in asymptomatic (AS) and symptomatic (S) stages. Although it has been speculated that post-symptomatic recovered individuals may be infectious for some period of time, this has not been considered in the model at this time due to lack of data. Hospitalised and critical individuals are assumed to be well isolated and also not able to infect others. The provided dynamic output of the reproduction number $R_t$ can be used to guide and interpret the impact of interventions in terms of $R_t$.

Modelled infected individuals can take only three possible infectious paths, namely: (i) AS → R; (ii) AS → S → R and (iii) AS → S → SH. These paths are made of combinations of four possible infectious stage intervals in which infected individuals spend time and infect at their corresponding rate (see Table 5).

The dynamic computation of $R_t$ consists of adding the total infection contributions of every stage interval as shown in Eq 6.

$$R_t = \Sigma[(r_{i\_asT}/N_{asT}) * (t_{r\_as}. * f_{r\_as} \quad + t_{s\_as}. * (1 - f_{r\_as})) + \\ (r_{i\_sT}/N_{sT}) * (t_{r\_s}. * (1 - f_{r\_as}). * f_{r\_s} + t_{sh\_s}. * (1 - f_{r\_as}). * (1 - f_{r\_s}))] \tag{6}$$

in which, the age group weighted average rates of infection by AS and S are given as per Eqs 7.

**Table 5. Possible infectious stages intervals for $R_t$ computation.**

| Infectious interval | Fraction of infected passing the interval ($ind_{interv}/ind_{inf}$) | Interval duration (d) | Total infections per stage interval per individual infected ($inf_{interv}/ind_{inf}$) |
|---|---|---|---|
| $AS \rightarrow R$ | $f_{r\_as}$ | $t_{r\_as}$ | $(r_{i\_asT}/N_{asT})^* \, t_{r\_as}.^* \, f_{r\_as}$ |
| $AS \rightarrow S$ | $(1-f_{r\_as})$ | $t_{s\_as}$ | $(r_{i\_asT}/N_{asT})^* \, t_{s\_as}.^* \, (1-f_{r\_as})$ |
| $S \rightarrow R$ | $(1-f_{r\_as})^* \, f_{r\_s}$ | $t_{r\_s}$ | $(r_{i\_sT}/N_{sT}) \, ^* \, t_{r\_s}.^* \, (1-f_{r\_as}).^* \, f_{r\_s}$ |
| $S \rightarrow SH$ | $(1-f_{r\_as})^* \, (1-f_{r\_s})$ | $t_{sh\_s}$ | $(r_{i\_sT}/N_{sT}) \, ^* \, t_{sh\_s}.^* \, (1-f_{r\_as}).^* \, (1-f_{r\_s})$ |

A and 7.B.

$$r_{i\_asT} = \Sigma(\mathbf{r_{i\_as}}.* \mathbf{N_{as}})/N_{asT}) \qquad [inf_{PS}/ind_{PS} \cdot d] \qquad (7.A)$$

$$r_{i\_sT} = \Sigma(\mathbf{r_{i\_s}}.* \mathbf{N_{s}})/N_{sT}) \qquad [inf_{S}/ind_{S} \cdot d] \qquad (7.B)$$

**Model limitations.** The model shares many of the fundamental characteristics of compartment SEIR-type models and is based on dynamic balances of individuals in compartments classified by their stage of infection and age groups only. No other differentiation within those groups is captured by the model. This characteristic allows for the model application only to single, densely populated clusters. The model has a low complexity and requires a small number of parameters that are also mechanistic and meaningful. Most of these parameters can be directly estimated from epidemiological and clinical data and are not recommended for calibration against data. Analogous to all SEIR-type models this model carries limitations since all variables and parameters refer to representative averages for each compartment of stage and age group. This may limit the model representation of the non-linear interactions that occur in reality. Phenomena like so-called super spread events or any location specific phenomena are not reproduced by these types of models. Any quantitative interpretation of results for prediction purposes should therefore be always accompanied by a critical discussion against these limitations.

**Table 6. Intervention parameters.**

| Definition | Parameter | Units |
|---|---|---|
| Average daily individual contacts by H | $\mathbf{ni_h}$ | #interactions / (H individual·day) |
| Personal protection and awareness by H[1] | $\mathbf{lpa_h}$ | Ø |
| Personal protection and awareness by PS[1] | $\mathbf{lpa_{as}}$ | Ø |
| Personal protection and awareness by S[1] | $\mathbf{lpa_s}$ | Ø |
| Likelihood of infection by interaction with PS[2] | $\mathbf{p_{i\_as}}$ | infections / interactions with AS |
| Likelihood of infection by interaction with S[2] | $\mathbf{p_{i\_s}}$ | infections / interactions with S |
| Percentage of tested individuals from PS[1] | $pt_{as}$ | #random (NI+AS) tested/#(H+NI+AS) |
| Percentage of tested individuals from S[1] | $pt_s$ | #random S tested/#S |
| Test sensitivity on AS | $t_{sns\_as}$ | #AS detected / #AS tested |
| Test sensitivity on S | $t_{sns\_s}$ | #S detected / #S tested |
| Reduction of interactions by unaware S[1] | $\mathbf{rfi_s}$ | Ø |

[1]Values only within the interval [0,1]

[2]Calculated, not an input parameter.

## Modelling interventions

The model presented above was used to evaluate different possible interventions to slow the spread of the COVID-19 outbreak. Four main interventions that have been or may be applied were evaluated namely (i) the *degree of social isolation* of the individuals in the population; (ii) the *level of personal protection and awareness* that individuals apply to protect themselves and others against contagion during interactions; (iii) the *implementation of mass testing* and (iv) the *intensive care capacity* available. These interventions can be stratified by age groups. The key parameters that define the interventions as modelled are described in Table 6.

**Social isolation.** The *degree of social isolation* is described through its impact on the parameter ($ni_h$) (vector per age group) corresponding to the representative average number of interpersonal contacts that healthy susceptible individuals have with others per day. Different $ni_h$ values can be applied to different age groups to describe age selective isolation strategies such as e.g. isolation of the elderly and/or the young alone.

**Level of protection and awareness.** The *level of protection and awareness* describes the use of personal and protective equipment (e.g. masks). The level of protection impacts the likelihood of infection (see Eq 4) and is described through the parameters ($lpa_h$) for healthy and ($lpa_{as}$ and $lpa_s$) for infectious AS and S individuals (both in vectors per age group). Values of the $lpa$ parameters can vary between 0 and 1, with 1 corresponding to the use of complete protective measures and zero to no use of any protection equipment. Analogously to the nih parameter, different values can be assigned per age group e.g. elders having a higher $lpa_h$ value due to being more prone to wearing masks compared to teenagers.

The likelihoods of infection per interaction are calculated as per Eqs 8.A and 8.B.

$$p_{i\_as} = (1 - lpa_h) .* (1 - lpa_{as}^{av}); \tag{8.A}$$

$$p_{i\_s} = (1 - lpa_h) .* (1 - lpa_s^{av}); \tag{8.B}$$

where $lpa_h$ reflects a level of protection and awareness related to interventions and defined below; $lpa_{as}^{av}$ and $lpa_s^{av}$ are scalars corresponding to the weighted averages over all age groups of the pool of AS and S with which H individuals can interact (Eqs 8.C and 8.D).

$$lpa_{as}^{av} = \Sigma(N_{as} .* lpa_{as})/N_{asT} \tag{8.C}$$

$$lpa_s^{av} = \Sigma(N_s .* lpa_s)/N_{sT} \tag{8.D}$$

where $N_{asT}$ and $N_{sT}$ are the total numbers of AS and S individuals of all ages respectively. The $\Sigma$ symbol indicates summation across all age groups.

**Awareness of infection by mass testing.** The *awareness of infection* after a positive test is assumed to lead to a full quarantine and removes those individuals from regular interaction with others. This *awareness of infection* is described through a reduction factor of interactions ($rfi$). Symptomatic individuals are already assumed to have a self-imposed quarantine ($rfi_s$) even if unaware of a positive test.

The fractions of individuals (without and with symptoms) aware of infection is therefore equal to the product of the fraction of total individuals randomly tested from the entire population ($pt_{as}$ for AS) and from the entire pool of individuals showing symptoms ($pt_s$ for S) times the corresponding test sensitivity ($t_{sns\_as}$ and $t_{sns\_s}$ respectively). Test sensitivity might differ between each of the two groups (symptomatic and asymptomatic). In asymptomatic and mild disease individuals tested, a sensitivity of the tests of 80% was assumed.

The fractions of infectious AS and S individuals that remain in interaction with others ($f_{ias}$ and $f_{is}$) are therefore calculated as per Eqs 8.E and 8.F. Hospitalised, critical and deceased are

considered excluded from the pool of interacting individuals.

$$f_{ias} = \Sigma[(1 - pt_{as} * t_{sns\_as}) * \mathbf{N_{as}}]/\Sigma[\mathbf{N_h} \cdot + \mathbf{N_{ni}} \cdot + (\mathbf{1} - pt_{as} * t_{sns\_as}) * \mathbf{N_{as}} \cdot + (1 - pt_s * t_{sns\_s}) * \mathbf{rfi_s} \cdot * \mathbf{N_s}) \cdot + \mathbf{N_r}] \tag{8.E}$$

$$f_{is} = \Sigma[(1 - pt_s * t_{sns\_s}) * \mathbf{rfi_s} \cdot * \mathbf{N_s}]/\Sigma[\mathbf{N_h} \cdot + \mathbf{N_{ni}} \cdot + (\mathbf{1} - pt_{as} * t_{sns\_as}) * \mathbf{N_{as}} \cdot + (1 - pt_s * t_{sns\_s}) * \mathbf{rfi_s} \cdot * \mathbf{N_s}) \cdot + \mathbf{N_r}] \tag{8.F}$$

where $pt_s$ is the proportion of symptomatic individuals tested (randomly) and $pt_{as}$ is the proportion of randomly tested non-symptomatic individuals of all types. The parameters $t_{sns\_s}$ and $t_{sns\_as}$ refer to the sensitivity of the tests for both groups, respectively. Table 3 shows the definitions and units of all the parameters used in the modelling of interventions.

**Critical care capacity.** The impact of available critical care capacity is modelled by a specific function to allocate critically ill individuals as per the available ICU. The function allocates critically ill individuals in two possible groups, namely those admitted to ICU ($\mathbf{N_{sc\_ic}}$) and those not admitted to ICU due to lack of capacity or for other medical reasons ($\mathbf{N_{sc\_ncc}}$). At each simulation time point the allocation function is computed for the total $\mathbf{N_{sc}}$ per age group.

The function allocates ICU resources with priority to age groups with higher ICU survival rate ($f_{r\_sc}$) until the maximum number of intensive care units is reached leaving any remaining individuals without care, in this way $\mathbf{N_{sc\_ic}}$ and $\mathbf{N_{sc\_ncc}}$ are computed.

As the COVID-19 outbreak has progressed, data indicate that not all patients in critical condition have been admitted into intensive care units (ICU). Data show that many individuals with very poor prognosis, particularly those of oldest age may have never been referred to ICU due to capacity limitations or other medical humanitarian reasons. This is based on data from Spain as of May 2020 [32] showing that for individuals over 70, only a fraction of the reported fatalities previously hospitalised was ever admitted. In order to maintain consistency with the reported data as of May 2020 [32] the parameters of $f_{d\_sc}$ and $f_{sc\_sh}$ have been estimated such that the product of $f_{d\_sc} * f_{sc\_sh}$ (fatality ratios over hospitalised individuals) is consistent with reported numbers for all ages irrespective of reported ICU admissions.

## Results and discussion

### Impact of interventions on a COVID-19 outbreak case study

A case study based on a scenario of propagation of the COVID-19 pandemic using data available from Spain as of May 2020 [32, 33] is presented below. The results obtained are intended to be interpreted qualitatively and to be contextualized to the specific setting characteristics. They serve also as a demonstration of the model potential if applied with higher confidence parameter values. Several selected scenarios were simulated aimed at illustrating the impact of different interventions.

Default reference epidemiological and clinical parameter values were extracted from different information sources on the COVID-19 outbreak as available in May 2020 [24, 32, 34, 35]. Details of values and sources are provided in S1 and S2 Tables in S1 File respectively, with indications of the level of confidence. A population with an age distribution as that of the region of Madrid (Spain) in 2019 was used [36] for the simulations.

Default reference values for interventions-related parameters (number of interactions per age group, level of protection awareness, and the isolation due to testing) were selected in order to describe a situation prior to the outbreak and without any specific intervention (see values and rationale in S3 Table in S1 File). The value for available intensive care beds per million people (**capICpM**) of 261 has been used by default in all case studies. The dynamic

simulation results of the default outbreak scenario under no intervention is shown in S4 Fig in S1 File.

All scenarios are simulated for 365 days and evaluated in terms of (i) final total number of fatalities at outbreak termination and (ii) final number of fatalities per age group. In addition, the scenarios are presented also in terms of dynamic profiles over time for (iii) number of active cases; (iv) reproduction number; (iv) number of critical cases; (v) number of fatalities.

**Intervention #1. Social isolation.** Under this scenario, the impact of the degree of social isolation ($ni_h$) was evaluated by age brackets. Four cases are shown, namely (i) universal social isolation, (ii) isolation for elders only (for individuals over 60 years old), (iii) isolation for youngsters (below 20 years old), and (iv) isolation for both elders and youngsters (only people 20 years or older and younger than 60 years old were not isolated).

The comparison between the impacts of the four types of isolations is shown in Fig 4. For the complete results of the four isolation intervention cases, please refer to the S1 File, Section VI.

It appears that social isolation starts to be effective in terms of significantly reducing total deaths after the number of daily contacts is placed below a threshold number (approximately two to three daily contacts in this example). Above this number of contacts per day, it does not appear to significantly modify the total final fatalities. This represents a decrease in number of contacts from 70–80%, which falls in line with the observations reported in literature that social distancing required over 80% to be effective [13].

Full details of the results shown in Fig 4 can be seen in (see S6 Fig in S1 File). The model is capable of capturing the effect of the number of interactions partly due to its description of the

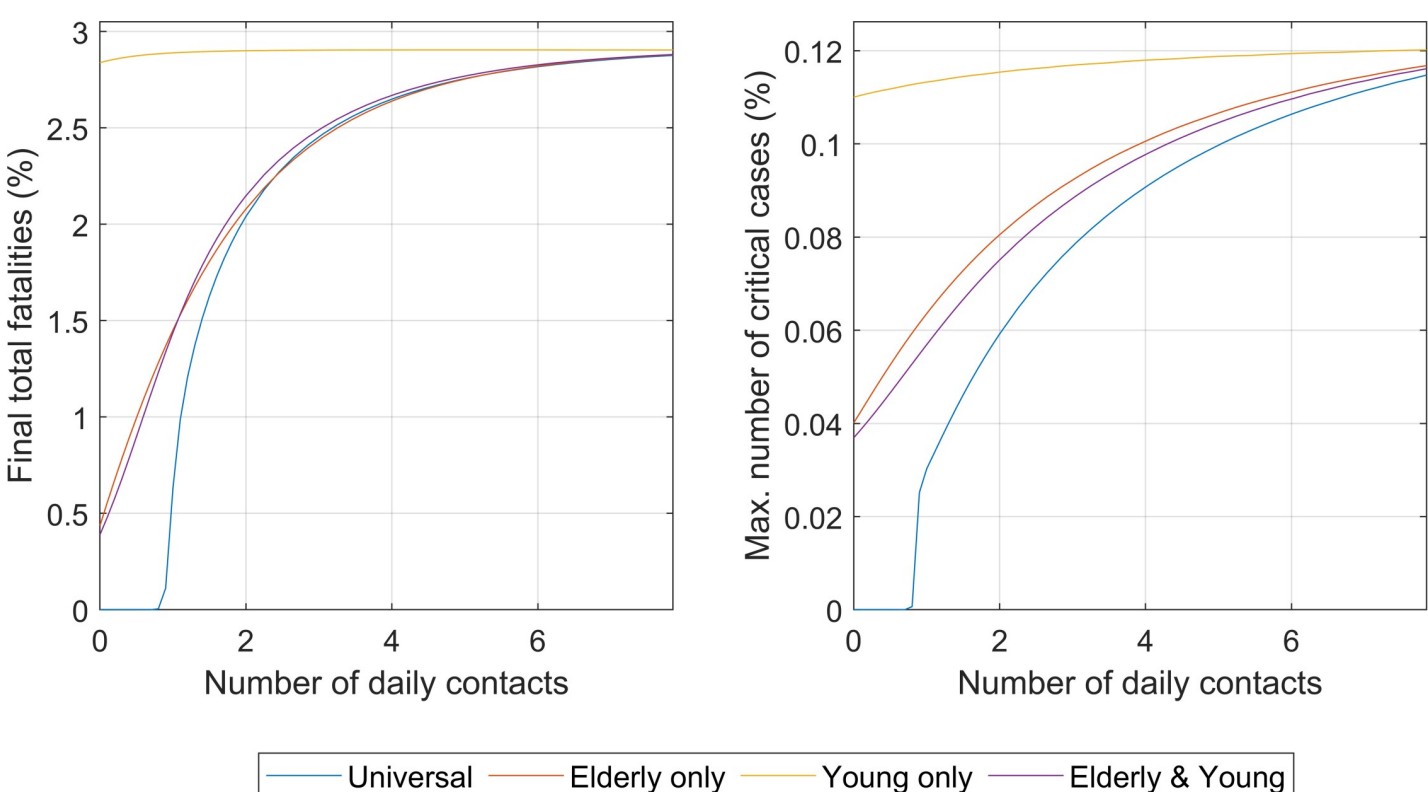

**Fig 4. Impact of the degree of social isolation under the four different strategies (universal and per age groups) on the total number of fatalities at the end of the outbreak (left) and on the maximum values of critically ill individuals ever reached during the outbreak (right).** Numbers are as percentage of the total population.

saturation of the healthcare capacity. S6a Fig in S1 File (middle left) illustrates the "flatten the curve" concept, now globally popular. In line with other publications [10, 11], if interactions are not decreased, the number of cases grows rapidly and explosively. S6a Fig in S1 File (middle right) shows the estimate of $R_t$ over time. This illustrates how the level of social isolation can define the infectability and the number of cases each infected individual will infect ($R_t$) showing how factors such as interventions can impact $R_t$. The number of critical cases increases throughout time as the social interaction increases S6a Fig in S1 File (bottom left).

As shown in Fig 4, the selective social isolation of the elderly has a potentially very significant impact on final total fatalities at an almost comparable level than for universal isolation. This is consequence of the much higher mortality by the disease in the elder than in the younger (S1 Table in S1 File, Section I) [32]. This is a result with potentially significant consequences as it indicates that a sustained isolation selective only to the elderly and not to the other age groups could be applied for economic and social reasons at a small cost in number of increased total fatalities. The isolation of the young individuals ($< 20$ years old) produces no effect in the overall number of total fatalities. The detailed results of imposed social isolation selective to age groups is shown in S6b-S6d Fig in S1 File.

Many of the early interventions during the COVID-19 outbreak started by isolating the elderly and the young (no schools, no colleges, or universities for students), decreasing the number of interactions of the two subpopulations substantially. The isolation of these population groups together results in similar effects to that of the isolation of the elderly alone with no significant added value in isolating the young respect only the elderly as shown in Fig 4.

**Intervention #2. Level of protection and awareness.**   The impact of the level of self-protection and awareness was evaluated in this scenario. The parameter that describes this intervention is the level of personal protection and awareness (**lpa**) of the healthy and infectious population groups (**lpa_h**, **lpa_as** and **lpa_s**) (see S3 Table in S1 File). Increases in these parameters decrease the likelihood of infection per interaction (see Eqs 8.A and 8.B) and subsequently the rates of infection (Eqs 4.A and 4.B). Fig 5 presents the main impacts of this intervention on the total final fatalities and on the maximum number of critical cases. For the full descriptive results, including the predictions of the effective reproduction number $R_t$, please refer to S7 Fig in S1 File, Section VII.

Fig 5 shows that high levels of protection and awareness appear as having a potentially major impact, leading to a very significant reduction on total outbreak fatalities at the highest levels of protection. This is in line of the substantial impact of the masks previously reported in other models [22, 37], however in their case they point that even low protection of masks might deem important decreases in number of cases. The height of the peak of critical cases decreases the higher the level of protection is, Fig 5 (right). In addition, if the peak does not exceed the critical care capacity (0.026% of the population), the total number of fatalities reaches much smaller values. Although moderate values for the levels of protection and awareness of 60% or lower decrease the speed of the outbreak they eventually yield the similar number of fatalities due to the exceeded ICU capacity.

**Intervention #3. Awareness of infection by testing.**   The impact of widespread extensive random testing was evaluated in this scenario. Fig 6 shows the comparison of the impact of testing only symptomatic individuals, testing randomly only the non-symptomatic population (therefore the same fraction of asymptomatic) and testing everyone both with and without symptoms.

Only when both symptomatic and asymptomatic groups are extensively tested to levels allowing for a detection of near 90% of total infections, a meaningful impact is predicted. The current test sensitivities (around 80%) imply that even at 100% of population tested those high levels of detections required will be unachievable and a large number of infections will remain

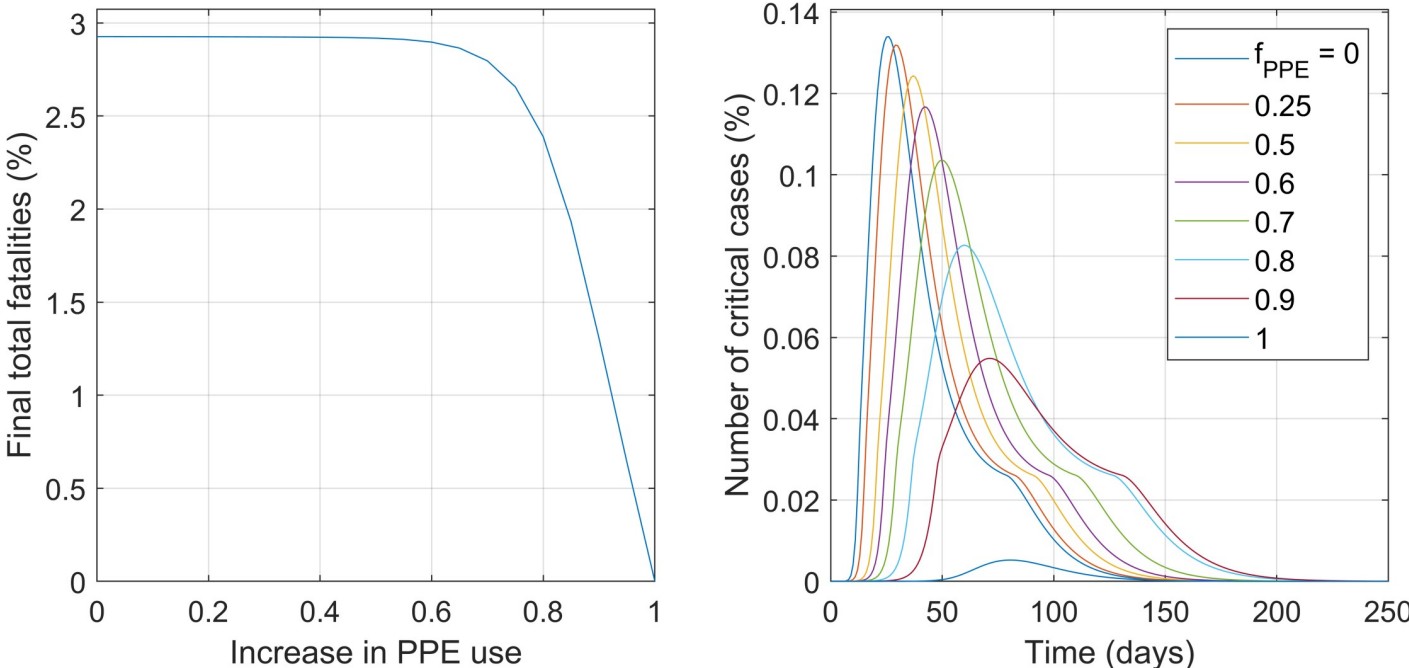

**Fig 5. Impact of the *levels of protection and awareness* on the total number of fatalities at the end of the outbreak (left) and on the maximum values of critically ill individuals ever reached during the outbreak (right).** Numbers are as percentage of the total population.

undetected. Therefore, the intervention appears to be ineffective if only applied by itself (for the complete simulation results of the impact of this intervention, please refer to S8a-S8c Fig in S1 File, Section VIII). The number of fatalities (and also number of cases) could be decreased substantially only when mass testing is combined with other interventions (social isolation, increased level of protection and/or increase in ICU beds) [11]. In addition, Fig 6 shows clearly that testing only symptomatic individuals leads to barely no impact at all. The results show that extensive testing as a sole strategy (not combined with other interventions such as social isolation, higher lpa and increase in ICU beds) appears as unfeasible to decrease the number of fatalities due to the high percentage of detections, sustained over time, required.

**Intervention #4. Critical care capacity.**   The impact of the availability of intensive care beds was evaluated in this scenario. The parameter that describes this intervention is the number of available intensive care beds per million population. Fig 7 illustrates the model predictions for this scenario, in terms of total final fatalities and numbers of critical cases over time. Note that once the ICU beds capacity is exceeded the critically ill patients become fatalities in one day.

From Fig 7 (for more complete details, refer to S9 Fig in S1 File, Section IX), the large impact that the increase in critical care resources can have in decreasing total fatalities becomes evident. The higher the availability of critical beds, the lower number of fatalities. These appears to be in accordance with the interventions in several countries for building field hospitals [38]. The trend applies until there is no shortage of IC beds and all remaining fatalities are only the unavoidable ones. This intervention avoids those deaths that are preventable by the availability of critical care support for those that need it.

## Conclusions

The impact of specific interventions on the outbreak time course, number of cases and outcome of fatalities were evaluated. Data available from Spain for the COVID-19 outbreak as of

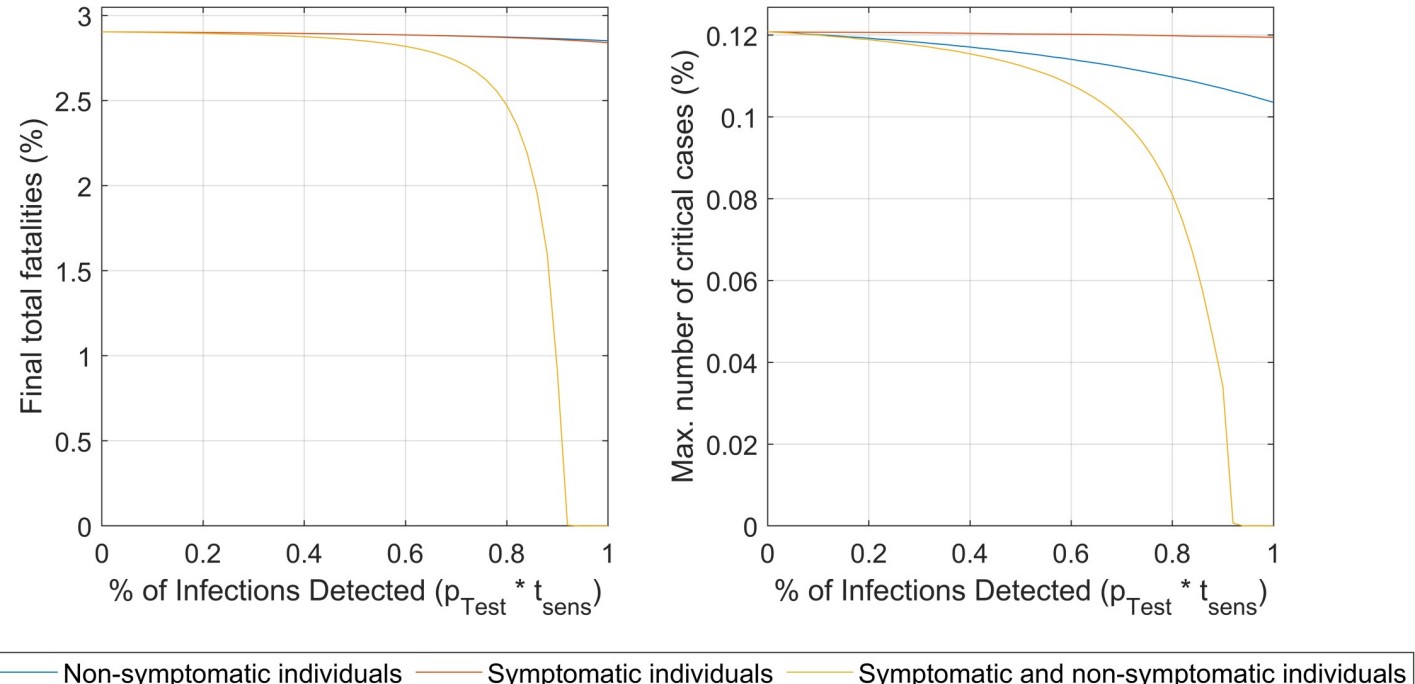

**Fig 6. Impact of the infection detection level (product of the proportion of individuals tested times the test the sensitivity, 80%) for symptomatic alone, asymptomatic alone and for both.** The impact on the total number of fatalities at the end of the outbreak (left) and on the maximum values of critically ill individuals ever reached during the outbreak (right) are shown. Numbers are as percentage of the total population.

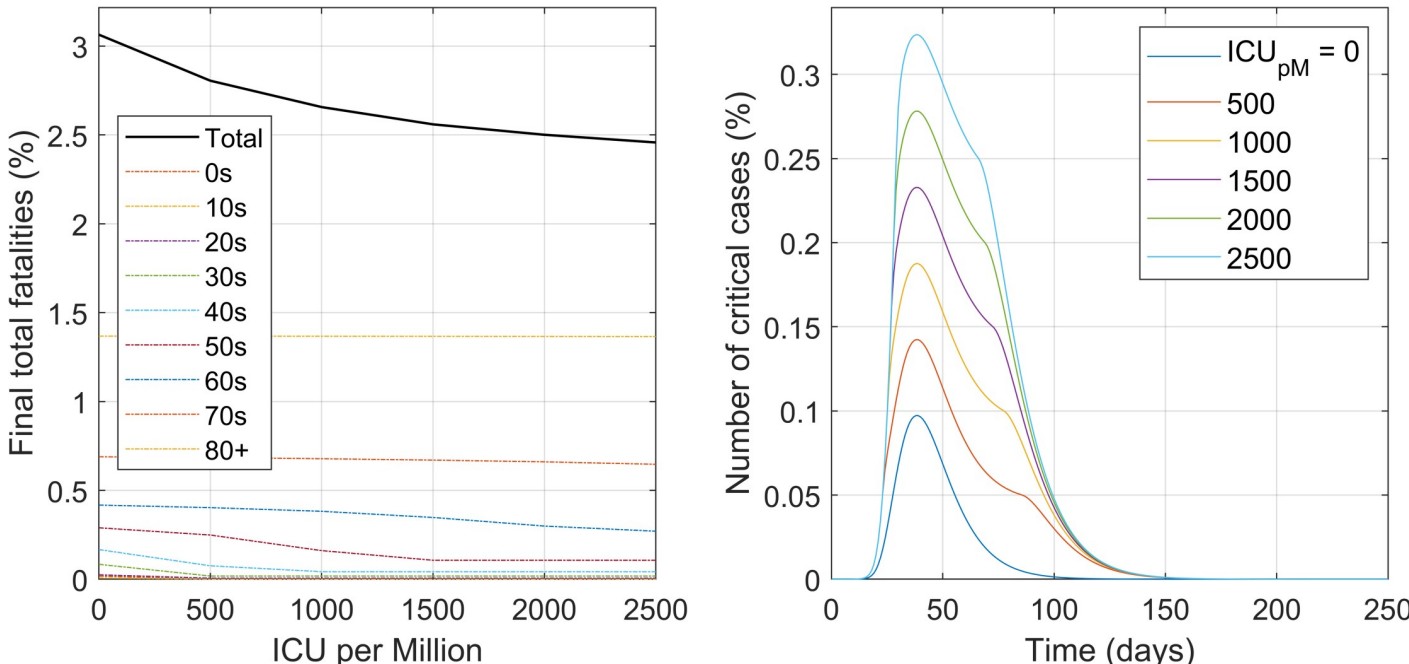

**Fig 7. Impact of the availability of intensive care beds on the final number of fatalities total and also per age group (left) as well as for the different time course profiles of the number of critical cases (right).** Numbers are in percentage of the total population of all ages.

May 2020 was used. Our results on the impact and mechanisms of the various interventions indicate that:

1. Universal social isolation measures may be effective in reducing total fatalities only if they are strict and the average number of daily social interactions is reduced below a low threshold number.

2. Selective isolation of only the age groups most vulnerable to the disease (i.e. older than 60) appears almost as effective as the universal isolation in reducing total fatalities but at a possible much lower economic damage. The compared impacts of different social isolation interventions on the final total number of fatalities (Fig 4, left) shows that the isolation of only the elderly can achieve equivalent impact to that of all.

3. An increase in the number of critical care beds could save a significant number of lives. Using our current parameters values, an estimate of 8 fatalities could be avoided per extra available critical care bed unit.

4. The use of protective equipment (PPE) appears capable of reducing very significantly the total fatalities if implemented extensively and to a high degree.

5. Extensive random testing of the population leading to isolation of the infected individuals, appears to be an ineffective intervention (if applied only by itself) due to the required (unreachable with existing test sensitivities) high percentage of infection detections and the logistic difficulty for its maintenance over time.

Any quantitative interpretation of the above results must be accompanied with a critical discussion in terms of the model limitations and its frame of application. The sensitivity analyses provided can be used to help such analysis.

## Supporting information

**S1 File. The Matlab® source code and Excel file containing all parameter values used as well as a non-age segregated version of the model are available at https://github.com/ EnvBioProM/COVID_Model.**
(PDF)

## Acknowledgments

All authors wish to thank Khalifa University and the Government of Abu Dhabi for the support.

## Author Contributions

**Conceptualization:** Juan M. Acuña.

**Data curation:** Jorge Rodríguez, Mauricio Patón.

**Formal analysis:** Jorge Rodríguez, Mauricio Patón, Joao M. Uratani, Juan M. Acuña.

**Funding acquisition:** Jorge Rodríguez, Juan M. Acuña.

**Investigation:** Jorge Rodríguez, Mauricio Patón, Joao M. Uratani.

**Methodology:** Jorge Rodríguez.

**Project administration:** Jorge Rodríguez.

**Resources:** Jorge Rodríguez, Juan M. Acuña.

**Software:** Jorge Rodríguez, Mauricio Patón.

**Supervision:** Jorge Rodríguez.

**Validation:** Jorge Rodríguez, Mauricio Patón.

**Visualization:** Jorge Rodríguez, Mauricio Patón.

**Writing – original draft:** Jorge Rodríguez, Mauricio Patón, Joao M. Uratani.

**Writing – review & editing:** Jorge Rodríguez, Mauricio Patón, Joao M. Uratani, Juan M. Acuña.

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
