## [Decision Letter · Decision Letter 0]

13 Nov 2020

PONE-D-20-26796

Modelling the impact of interventions on the progress of the COVID-19 outbreak including age segregation

PLOS ONE

Dear Dr. Rodríguez R.,

Thank you for submitting your manuscript to PLOS ONE. After careful consideration, we feel that it has merit but does not fully meet PLOS ONE’s publication criteria as it currently stands. Therefore, we invite you to submit a revised version of the manuscript that addresses the points raised during the review process.

We look forward to receiving your revised manuscript.

Kind regards,

Rashid Ansumana

Academic Editor

PLOS ONE

Journal Requirements:

2. Please ensure that you refer to Figure 3 in your text as, if accepted, production will need this reference to link the reader to the figure.

3. Please upload a copy of Figure 11, to which you refer in your text on page 25. If the figure is no longer to be included as part of the submission please remove all reference to it within the text.

4. Please upload a copy of Supporting Information Figure A2 which you refer to in your text on page 17.

5. Please include a copy of Table 5 which you refer to in your text on page 13.

Reviewers' comments:

Reviewer's Responses to Questions

**Comments to the Author**

1. Is the manuscript technically sound, and do the data support the conclusions?

Reviewer #1: No

Reviewer #2: Partly

2. Has the statistical analysis been performed appropriately and rigorously? 

Reviewer #1: No

Reviewer #2: N/A

3. Have the authors made all data underlying the findings in their manuscript fully available?

Reviewer #1: No

Reviewer #2: Yes

4. Is the manuscript presented in an intelligible fashion and written in standard English?

Reviewer #1: Yes

Reviewer #2: Yes

5. Review Comments to the Author

Reviewer #1: The authors addressed an interesting subject of global health importance. COVID-19 interventions were evaluated through a mechanistic model. The authors compartmentalized age groups to assess impact of COVID-19 interventions on the dynamics of spread of the disease, and the final size of the epidemic.

I commend the authors for their efforts. However, I have some major concerns:

1. The Abstract is too long and unfocused; same goes for the Introduction section. Most of the information in the introduction are too generic with no direct relevance to COVID-19.

The study is assesses the impact of different interventions on COVID-19 spread, so, one would expect the introduction section to include a brief discussion of the literature on modelling studies that have done similar investigation.

3. The model development is confusing and the notations are not intuitive - too many subscripts in some of the state variables and parameters. For instance, notations such as rsh_s will be difficult to follow.

4. Some of the model assumptions do not make practical/biological sense. For example, the model assumes that all critical cases without access to ICU will result in fatality. This is not justifiable, and it will obviously overestimate the fatality rate. Additionally, presymptomatic cases, by definition, will present with symptoms at some point in the future. The model assumes that presymptomatic individuals can recover without intervention.

5. The model and result discussion are highly speculative. Very few references were made to previous works on COVID-19 modelling in the Discussion. Moreover, the results do not support most of the claims in the discussion section.

6. The flaws in the model development and assumptions became more apparent in the results: in figure 4 for example, it is strange that the % of fatality is higher than % of critical cases. This is clearly wrong, since fatality is a subset of critical cases. Moreover, it is also suspicious that you have 0 critical cases if the number of contact per day is about 0.9. Additionally, 50% increase in PPE has no effect on the final total fatality? 90% of case detection will lead to 0 fatality? These can’t be right.

Reviewer #2: The authors have modeled the spread of a Covid-like infection across a population, the population being divided into 9 age groups from infancy to old age and the disease stage depicted from asymptomatic and pre-symptomatic to deceased. Equations are set up for transitions across disease stage in time. Real data was used to evaluate the parameters in the model. It is not clear that interventions such as isolation etc. were modeled as stated. It is claimed that the model is deterministic, but it is possible that there could be chaotic outcomes especially with multiple interacting equations.

To be reconsidered for publication

(1) the content can be made more crisp and clear.

(2) the length reduced reduced by about half

(3) bring out the conclusions and interpretation more clearly

(4) summarize the conclusions making reference to the claims set forward in the abstract

6. PLOS authors have the option to publish the peer review history of their article (what does this mean?). If published, this will include your full peer review and any attached files.

Reviewer #1: No

Reviewer #2: No

---

## [Author Response · Author response to Decision Letter 0]

8 Dec 2020

See attached file with all the detailed responses.

---

## [Decision Letter · Decision Letter 1]

17 Feb 2021

PONE-D-20-26796R1

Modelling the impact of interventions on the progress of the COVID-19 outbreak including age segregation

PLOS ONE

Dear Dr. Rodríguez R.,

Thank you for submitting your manuscript to PLOS ONE. After careful consideration, we feel that it has merit but does not fully meet PLOS ONE’s publication criteria as it currently stands. Therefore, we invite you to submit a revised version of the manuscript that addresses the points raised during the review process.

We look forward to receiving your revised manuscript.

Kind regards,

Martial L Ndeffo Mbah, Ph.D

Academic Editor

PLOS ONE

Additional Editor Comments (if provided):

Responses to the initial reviewers comments are satisfactory. Please, address the minor concerns raised by the reviewer before the manuscript can be deem suitable for publication.

Reviewers' comments:

Reviewer's Responses to Questions

**Comments to the Author**

1. If the authors have adequately addressed your comments raised in a previous round of review and you feel that this manuscript is now acceptable for publication, you may indicate that here to bypass the “Comments to the Author” section, enter your conflict of interest statement in the “Confidential to Editor” section, and submit your "Accept" recommendation.

Reviewer #3: (No Response)

2. Is the manuscript technically sound, and do the data support the conclusions?

Reviewer #3: Yes

3. Has the statistical analysis been performed appropriately and rigorously? 

Reviewer #3: N/A

4. Have the authors made all data underlying the findings in their manuscript fully available?

Reviewer #3: Yes

5. Is the manuscript presented in an intelligible fashion and written in standard English?

Reviewer #3: Yes

6. Review Comments to the Author

Reviewer #3: (No Response)

7. PLOS authors have the option to publish the peer review history of their article (what does this mean?). If published, this will include your full peer review and any attached files.

Reviewer #3: No

---

## [Author Response · Author response to Decision Letter 1]

21 Feb 2021

See attached file with detailed responses

---

## [Editor Report · Decision Letter 2]

23 Feb 2021

Modelling the impact of interventions on the progress of the COVID-19 outbreak including age segregation

PONE-D-20-26796R2

Dear Dr. Rodríguez R.,

We’re pleased to inform you that your manuscript has been judged scientifically suitable for publication and will be formally accepted for publication once it meets all outstanding technical requirements.

Kind regards,

Martial L Ndeffo Mbah, Ph.D

Academic Editor

PLOS ONE
---

## [Editor Report · Acceptance letter]

5 Mar 2021

PONE-D-20-26796R2 

Modelling the impact of interventions on the progress of the COVID-19 outbreak including age segregation 

Dear Dr. Rodríguez:

I'm pleased to inform you that your manuscript has been deemed suitable for publication in PLOS ONE. Congratulations! Your manuscript is now with our production department. 

Kind regards, 

on behalf of

Dr. Martial L Ndeffo Mbah 

Academic Editor

PLOS ONE